# Grain Chalkiness Is Decreased by Balancing the Synthesis of Protein and Starch in Hybrid Indica Rice Grains under Nitrogen Fertilization

**DOI:** 10.3390/foods13060855

**Published:** 2024-03-11

**Authors:** Changchun Guo, Lin Zhang, Peng Jiang, Zhiyuan Yang, Zongkui Chen, Fuxian Xu, Xiaoyi Guo, Yongjian Sun, Jun Ma

**Affiliations:** 1Key Laboratory of Southwest Rice Biology and Genetic Breeding, Ministry of Agriculture and Rural Affairs, Rice and Sorghum Research Institute, Sichuan Academy of Agricultural Sciences, Deyang 618000, China; changchuns1991@163.com (C.G.); zhanglin1090@sina.com (L.Z.); jiangyipeng137@163.com (P.J.); xu6501@163.com (F.X.); guoxiaoyi9@163.com (X.G.); 2Crop Ecophysiology and Cultivation Key Laboratory of Sichuan Province, Rice Research Institute, Sichuan Agricultural University, Chengdu 611130, China; dreamislasting@163.com (Z.Y.); chenzongkui90@foxmail.com (Z.C.); yongjians1980@163.com (Y.S.)

**Keywords:** N rate, chalkiness features, C and N metabolism, endogenous hormones, dynamic accumulation

## Abstract

The important reason for the commercial value of hybrid rice suffering is due to excessive chalkiness, and the biosynthesis of starch and proteins is critical for regulating chalkiness; however, it is currently unclear how the application of N fertilizer affects grains to reduce their chalkiness and improve their quality. The 2019, 2020, and 2021 trials were conducted in a split-plot design, with high and low chalky varieties as the main plot and N fertilizer rate as the split-plot. The effects of fertilization with 75, 150, and 225 kg N ha^−1^ on the dynamic synthesis of starch, protein, and endogenous hormones and on the amino acid of hybrid indica rice kernels with different degrees of chalkiness were investigated. Grain physiological activity was higher in low-chalky varieties than in high-chalky varieties, and these physiological parameters were strongly associated with chalkiness formation. Higher N fertilization (150 and 225 kg N ha^−1^) significantly reduced the proportion of chalky grains (8.93–28.02%) and chalkiness (8.61–33.99%) compared with 75 kg N ha^−1^. Increased N fertilization decreased the activities of granule-bound starch synthase and starch-debranching enzyme, but significantly increased adenosine diphosphate glucose pyrophosphorylase, soluble starch synthase, and starch-branching enzyme activities, synergistically improving glutamate synthetase and glutamine synthetase enzyme activities, which tended to support the synthesis of amylopectin, α-ketoglutarate, and 3-phosphoglyceric acid-derived amino acids in the endosperm cells of the grains; this favored starch and protein accumulation in the grains at 6–30 days after anthesis. Additionally, N application promoted the synthesis of endogenous hormones 1-aminocyclopropane-1-carboxylic acid, gibberellins, and abscisic acid in grains. Hence, N fertilization reduced the rice chalkiness in hybrid indica rice varieties by balancing grain protein and starch composition and enhancing some endogenous hormone synthesis.

## 1. Introduction

Continuous changes in the demographic structure and standard of living of modern societies have promoted the importance of rice quality over that of quantity [1,2]. Rice quality parameters include milling (whole brown and head rice rate), appearance (chalky grain rate and chalkiness degree), nutritional value (starch, protein, fat, and amino acid), and cooking quality (gel consistency, hardness, and viscosity) [3,4,5]. The southwestern hybrid rice planting area of China is among the top-producing regions in terms of grain yield; however, factors such as weak light, high humidity, and small temperature differences between day and night lead to a pronounced variability in terms of high quality, and the consistent production of high-quality rice is difficult. High chalkiness is a key factor limiting this overall quality [6]. Chalkiness is not only an important indicator for evaluating the appearance quality of rice, but is also a key factor limiting the economic efficiency of rice [7,8].

Rice chalkiness is primarily attributed to underdeveloped amyloplasts and proteins in the endosperm and to increased space between loosely arranged starch granules [9]. A high starch content of grain under high-temperature stress does not increase chalkiness [10], whereas an insufficient accumulation of protein between starch particles does [11]. The increase in proteosomes fills the gaps between the starch granules, which then reduces the chalkiness of rice [12]. Thus, starch plays a major role in regulating chalkiness, whereas proteins play only an auxiliary regulatory role. The metabolism of rice grain carbon (C) mainly relies on glycolysis, the tricarboxylic acid cycle, the pentose phosphate pathway, the glycolic acid oxidation pathway, and the glyoxylate cycle, whereas nitrogen (N) metabolism mainly involves the reduction in and assimilation of nitric acid and the synthesis of amino acids [13]. C metabolism through the tricarboxylic acid cycle is important to provide a C skeleton and energy for N metabolism [14], and N metabolism is crucial for C metabolism to facilitate the synthesis of key enzymes [15]. Thus, the two categories of metabolism constrain but also promote each other; their respective end products are starch and protein, accounting for approximately 75–85% and 6.5–9.0%, respectively, of the dry weight of brown rice. Consequently, maintaining an adequate C metabolism in rice grains promotes the N metabolism to synthesize proteins, achieve a C–N balance, and reduce chalkiness.

The synthesis of proteins and starch and the accumulation patterns of composite starch granules depend on the genotype, source–sink supply relationships, grain filling stage, endogenous hormone accumulation, and the synthesis of primary and secondary inclusions in the C and N metabolism pathways [16,17]. These factors are key to chalkiness formation, either directly or indirectly. N application is an essential cultivation measure for regulating rice grain filling, endogenous hormone accumulation, and C and N metabolism. High N fertilization affects the grain filling balance, leading to slower grain filling and poor filling, resulting in a loose starch arrangement and increased chalkiness [18]; however, some studies have suggested that increasing N fertilization can prolong the effective filling period of inferior grains and reduce chalkiness formation [19,20]. The activities of cytokinins (ZRs), indole acetic acid (IAA), abscisic acid (ABA), gibberellin (GA), soluble starch synthase (SS), and starch-branching enzyme (BE) within the grain are strongly correlated with chalkiness during the pre-filling period, whereas the opposite is true during the post-filling period [21]. The involvement of reactive oxygen species in the induction of α-amylase in rice kernels under heat stress promotes kernel maturation and leads to increased chalkiness in rice grains by regulating the GA/ABA metabolism [21]. Further, storage proteins, amino acids, and starch accumulation synergistically modulate high-temperature stress-induced chalkiness [16]. N application increases the activities of the enzymes associated with N metabolism under high-temperature stress, which, in turn, increases the protein content of rice grains and leads to a decrease in chalkiness; however, the difference in starch content is negligible [22], indicating that chalkiness is regulated by the co-regulation of the starch and protein in the grains.

Amino acids are precursors in protein biosynthesis, and their content, composition, and relative abundance can affect the chalkiness of rice. The levels of phosphoenolpyruvic acid and oxaloacetate group amino acids, such as phenylalanine, aspartic acid, and threonine, are markedly higher in transparent rice grains than in chalky rice grains [23]. Glutamic acid and aspartic acid account for high proportions of the amino acid composition of protein synthesized in rice grains, whereas lysine and threonine account for low proportions, but are considered as important amino acids that constrain protein synthesis. Furthermore, an increase in lysine and threonine in rice grains contributes to a high nutritional quality of rice and ensures that glutamic acid metabolism and tryptophan metabolism support the synthesis of glutelin and prolamin; thus, the lysine and threonine metabolism pathways are improved to allow for the synthesis of albumin and globulin [24]. Glutamate is the major component of rice grain glutelin, and the key enzymes involved in its synthesis are glutamine synthetase (GS) and glutamate synthetase (GOGAT) [25]. Increased GS and GOGAT enzyme activities in rice leaves and grains under N fertilizer regulation elicit an increase in glutamate and glutenin, which affects the chalky white quality.

N is an essential nutrient for rice growth, an important component of cellular nucleic acids, endogenous hormones, phospholipids, and proteins, and plays an irreplaceable role in its growth process. Appropriate N fertilizer application is conducive to maintaining the balance between C and N metabolism in rice plants and smooth grain filling, which reduces rice chalkiness [26], whereas excessive N abundance and N deficiency lead to increased chalkiness [27]. Under N deficiency, rice plants exhibit a poor nutrient supply, slower material transport and synthesis, insufficient grain filling, an increased number of empty grains, and an increased chalky grain rate and chalkiness. Under excessive N application, nutrient synthesis in rice plants and the grain filling rate are accelerated and grain filling fluctuates, which is not conducive to the accumulation of assimilates and grain filling, and is more likely to cause increased chalkiness [15,16].

Previous studies on the physiological mechanism of chalkiness formation predominantly focused on the key enzyme activities of grain C metabolism and differences in the accumulation and granule arrangement of the final starch product; however, the coordinated mechanisms of protein and starch biosynthesis in chalkiness formation in hybrid rice under N application remain unclear. We thus hypothesized that rice chalkiness may be effectively reduced by increasing endogenous hormones, the activities of key enzymes in C and N metabolism, and amino acid interactions in grains, thus balancing the biosynthesis of protein and starch under appropriate N fertilization. In this study, we conducted a three-year field experiment on hybrid indica rice varieties with varying chalkiness, which were subjected to three N fertilization schemes, in order (i) to test the effect of N application on the chalky characteristics of different rice varieties; (ii) to assess the accumulation of grain assimilates, endogenous hormones, and the activities of key enzymes of C and N metabolism, as well as the relationship between rice starch, protein, and amino acid composition and chalkiness formation; and (iii) to systematically analyze the response mechanism of rice chalkiness to N fertilization. Our findings provide a theoretical and practical basis for the production of low-chalkiness rice.

## 2. Materials and Methods

### 2.1. Study Sites and Plant Material

The experiment was established at Huihe Villag (30°43′ N, 103°47′ E), in Wenjiang District, Chengdu City, Sichuan Province, China, during the rice growing seasons in 2019–2021. The soil type was sandy loam; the soil basal nutrient content in the arable layer (0–20 cm) and meteorological information for the period from rice sowing to harvesting are shown in Table 1 and Figure 1. Two chalky white differential-type hybrid indica rice varieties, Chuannongyou 508 (Chuannong3A × Shuhui508, C1) and Shuangyou 573 (Shuang1A × Shuhui573, C2), which are representative rice varieties grown in southwest China, were used.

### 2.2. Experimental Design and Field Management

The trial was conducted in a split-plot design, with variety as the main plot and N fertilizer rate as the split-plot. Six treatments with three replications were used, totaling eighteen plots. The plot area was 5 m × 3 m. Simulating the mechanical precision hole direct seeding method for sowing, the two varieties were sown at the same time in each of the 3 years; C1 was sown on April 19 and C2 was sown after a 7-day interval. The row and hole spacing was 25 cm × 20 cm, with 300 holes per plot. After the seeds germinated and were exposed, four to six seeds were sown in each hole, and two seedlings were evenly fixed per hole after emergence.

The three N application schemes were 75, 150, and 225 kg N ha^−1^ (termed N1, N2, and N3, respectively). N was applied three times during the entire growth period: 40% at basal, 30% at tillering, and 30% at the panicle initiation stage. Phosphate (P_2_O_5_) and potash (K_2_O) fertilizers were applied at 75 and 150 kg ha^−1^, respectively: the P_2_O_5_ fertilizer was applied only as a basal application and the K_2_O fertilizer was applied in equal parts according to the basal fertilizer and spike fertilizer [28]. Ridges were established between plots and covered with plastic film to prevent water and fertilizer from leaking between the plots and to prevent and control pests, diseases, and weeds in the field.

### 2.3. Experimental Conditions and Procedures

#### 2.3.1. Sample Preparation

At the heading stage, 400 single spikes with the same spike size and flowering time were selected from each plot and marked with red tags. Fifteen single spikes were collected from these plots on days 6, 12, 18, 24, 30, 36, and 42 after anthesis. The first, second, and third grains from the base of the first and second branches on the spike were picked, heated at 105 °C for 45 min, dried at 80 °C until constant weight, hulled, and pulverized in a pulverizer, after which, they were passed through a 100-mesh sieve to conduct a dynamic determination of amylose, amylopectin, total starch, free amino acids, and proteins. Another 15 rice spikes were selected and wrapped in tin foil, after which, they were placed in liquid nitrogen for 30 min and stored at −80 °C. Before the assay, fresh grains from the same locations as described above were picked and skinned on ice trays, after which, they were placed in centrifuge tubes to measure the key enzyme activities of C and N metabolism, as well as endogenous hormones [29].

#### 2.3.2. Dynamic Analysis of Key Enzyme Activities in C Metabolism

The enzymatic activities of adenosine diphosphate glucose pyrophosphorylase (ADPG), granule-bound starch synthase (GBSS), SS, BE, and starch-debranching enzyme (DBE) were determined as described by Schaffer and Petreikov [30], Nakamura et al. [31], and Zhang et al. [32], with slight modifications and using biochemical kits. One unit of enzyme activity of ADPG, GBSS, SS, and BE indicated the amount of enzyme causing an increase in absorbance by one unit per g fresh weight per minute. One unit of DBE enzyme activity was defined as an increase of 1.0 mg in reducing sugar per minute. Twenty-five uniformly sized kernels were shelled, weighed 0.1 g as a sample, ground in 4.0 mL of extract (different for each enzyme), and then centrifuged at 11,000× *g* for 20 min, and the supernatant was collected to measure the ADPG, SS, BE, and DBE enzyme activities. The remaining precipitate was resuspended in 1.0 mL of the extraction solution for the determination of GBSS enzymatic activity.

For ADPG enzyme activity, 50 μL of supernatant was added to 1.0 mL of reaction mixture containing 100 mM Hepes-NaOH (pH 7.5), 5.0 mM DTT, 2.0 mM MgCl_2_, 0.5 mM NAD, 0.02 mM glucose 1,6-bisphosphate, 1.0 mM ADPG, 2.0 U phosphoglucomutase, and 2.0 U glucose 6-phosphate dehydrogenase. The reaction mixture was incubated at 30 °C for 5 min, followed by the addition of 1.0 mM PPi to initiate the reaction. The absorbance was measured at 340 nm for 5 min.

For the SS and GBSS activities, 15 mM DTT and 50 μL of resuspended precipitate were added to 280 μL of reaction mixture with 50 mM Hepes-NaOH (pH 7.4), 1.6 mM ADP, and 0.7 mg branched-chain starch. After boiling for 0.5 min in a water bath to inactivate the enzyme, 100 μL of a mixed solution containing 200 mM KCl, 50 mM Hepes-NaOH (pH 7.4), 10 mM MgCl_2_, 4.0 mM PEP, and 1.2 U pyruvate kinase was added, followed by incubation for 30 min at 30 °C. Thereafter, the mixture was incubated in a boiling water bath for 5 min, followed by centrifugation at 10,000× *g* for 5 min, and 300 μL of supernatant was mixed with 300 μL of reaction solution containing 50 mM Hepes-NaOH (pH 7.4), 20 mM MgCl_2_, 10 mM glucose, and 2.0 mM NADP. Finally, absorbance at 340 nm was measured immediately after the addition of hexokinase and 0.35 U glucose 6-phosphate dehydrogenase.

For BE activity, 50 μL of supernatant was added to 200 μL of reaction mixture containing 50 mM Hepes-NaOH (pH 7.4), 5.0 mM glucose 1-phosphate, 1.25 mM AMP, and 54 U phosphorylase. The reaction was terminated by adding another 50 μL of 1.0 M HCl, after which, the solution was mixed with 300 μL of dimethyl sulfoxide and 700 μL of 1.0% KI and 0.1% I_2_, and absorbance was measured at 540 nm.

For DBE activity, 100 μL of supernatant was added to 1.0 mL of reaction mixture containing 100 mM Hepes-NaOH (pH 7.5) and 2.0 mg pullulan, followed by incubation at 30 °C for 30 min. The reaction was terminated by incubation in a boiling water bath for 3 min. Finally, 0.83 mL if 3,5-dinitrosalicylic acid was added and the mixture was incubated in a boiling water bath for 10 min, after which, 3.7 mL of deionized water was added, and absorbance was measured at 540 nm.

#### 2.3.3. Dynamic Analysis of Key Enzyme Activities in N Metabolism

The enzymatic activities of glutamate dehydrogenase (GDH), GS, and GOGAT were determined according to Xi et al. [15] and Jia et al. [33], with slight modifications and using biochemical kits. The enzyme activity units of GDH and GOGAT were defined as 1 nmol of NADH consumed per min per gram fresh weight. One unit of GS enzyme activity of GS is expressed as a change in the absorbance value of 0.005 per min per gram fresh weight. Fresh grain samples (0.1 g) were placed in a mortar with 1 mL of extract (pH 8, 0.05 M Tris-HCl, 2 mM MgSO_4_, 2 mM DTT, and 0.4 M sucrose) and were ground to homogenization in an ice bath. The mixture was centrifuged at 10,000× *g* and 4 °C for 10 min, and the supernatant was collected for the assays of the GS, GOGAT, and GDH activities.

For GS activity, 150 μL of supernatant was added to 300 μL of reaction mixture containing 100 mM KH_2_PO_4_ buffer, 450 mM MgSO_4_·7H_2_O, 300 mM hydroxylamine, 100 mM ATP, 4.0 mM EDTA, and 1.0 M sodium glutarate, followed by incubation at 37 °C for 30 min; 150 μL of FeCl_3_ reagent was added to a blank control instead of the supernatant. Subsequently, the reaction was terminated by adding 500 μL or FeCl_3_ and centrifugation at 8000× *g* and 4 °C for 10 min. Finally, absorbance was measured at 540 nm.

For GOGAT activity, 200 μL of supernatant was added to a reaction mixture containing 1.8 mL of 25 mM Tris-HCl, 0.5 mL of 0.1 M α-ketoglutaric acid, 0.4 mL of 20 mM L-glutamine, 0.2 mL of 3.0 mM NADH, and 0.1 mL of 10 mM KCl at a final volume of 3.2 mL. The reaction was initiated by the addition of L-glutamine and NADH after enzyme preparation. A decrease in absorbance was recorded at 340 nm for 5 min.

For GDH activity, 0.5 mL of the supernatant was added to a reaction mixture containing 0.4 mL of 25 mM Tris-HCl buffer (pH 8.0), 0.2 mL of 20 mM L-glutamine, 0.15 mL of 0.1 M 2-oxoglutarate, 0.15 mL of 1.0 M NH_4_Cl, and 0.1 mL of 3 mM NADH at a final volume of 3.2 mL. The reaction was initiated by the addition of L-glutamine and NADH after enzyme preparation. The decrease in absorbance at 340 nm was recorded for 3 min.

#### 2.3.4. Dynamic Analysis of Endogenous Hormones

A crushed sample (0.1 mg) was added to 1 mL of ice-cold 50% aqueous acetonitrile. The samples were sonicated at 4 °C for 3 min and then extracted for 30 min. After centrifugation at 16,000× *g* and 4 °C for 10 min, the supernatant was collected and passed through an RP-SPE column: 1 mL of 100% methanol and 1 mL of deionized water were added separately, followed by equilibration with 50% aqueous acetonitrile. After loading the sample, the column was rinsed using 1 mL of 30% acetonitrile, and the components were collected. Thereafter, the samples were evaporated to dry in a stream of nitrogen, after which, they were dissolved in 200 μL of 30% acetonitrile and transferred to a sample vial containing the insert. 1-aminocyclopropane 1-carboxylic acid (ACC), ABA, IAA, GAs, and ZRs were determined using ultra-high-performance liquid chromatography (UPLC; Vanquish, Thermo Fisher Scientific, Waltham, MA, USA) and high-resolution mass spectrometry (Q Exactive, Thermo Fisher Scientific, Waltham, MA, USA). A Waters HSS T3 (Waters, Milford, Waltham, MA, USA) was used as the liquid chromatographic column with an injection volume of 2 μL, a column temperature of 40 °C, and a flow rate of 0.3 mL min^−1^. Electrospray ionization was used with an ion spray voltage of −2800 V. The scanning mode was single-ion detection mode with negative ions [34,35].

#### 2.3.5. Dynamic Analysis of Assimilates

Total starch, amylose, amylopectin, and protein were assayed as reported previously [12]. The determination of free amino acid content of rice kernels was conducted using a ninhydrin colorimetric method [36].

#### 2.3.6. Determination of Grain Chalkiness

At harvest, 3 kg of rice was taken from each plot and air-dried for 2 months. When the physicochemical properties of the grains were stabilized, it was dehulled and milled, and whole polished rice was obtained. Then, 1000 whole polished rice grains were randomly selected from each sample and placed on a scanner to create digital images and to assess the chalky grain percentage and chalkiness using an image analysis software (JMWT-12, Dongfujiuheng Instrument Technology Co., Ltd., Beijing, China) [19].

#### 2.3.7. Determination of Protein Fractions and Amino Acid Composition

Albumin, globulin, prolamin, and glutelin levels were assayed using the methods described by Lan et al. [37] and Zhang et al. [38], with minor modifications. The different protein fractions in the rice were separated and extracted sequentially with distilled water, 0.6 M NaCl, 70% ethanol, and 0.1 M NaOH. The content of each protein fraction in rice was quantitatively assessed using the Coomassie brilliant blue method.

The amino acid composition was determined using UPLC (Vanquish, Thermo Fisher Scientific, Waltham, MA, USA) and high-resolution mass spectrometry (Q Exactive, Thermo Fisher Scientific, Waltham, MA, USA). We weighed precisely 0.1 g of the sample into a centrifuge tube, added 0.5 mL of 0.1 M hydrochloric acid, vortexed the mixture, and incubated it for 1 h at room temperature, followed by centrifugation at 16,000× *g* for 10 min. Then, we collected 10 µL of the supernatant, placed it in a test tube, and added 70 µL of AccQ-Tag Ultra Borate buffer and 20 µL of AccQ-Tag reagent (Waters). The reaction mixture was heated to 55 °C for 10 min, cooled, and assayed using a reader. A BEH C18 liquid chromatographic column (Waters) was used with an injection volume of 1 μL, a column temperature of 55 °C, and a flow rate of 0.5 mL min^−1^. Electrospray ionization was performed at an ion spray voltage of +3000 V. The scanning mode was the FullScan mode with positive ions.

#### 2.3.8. Statistical Analysis

Analysis of variance was performed using SPSS (version 22.0; SPSS Inc., Chicago, IL, USA). Differences between years, periods, and varieties were tested using a least significant difference test at *p* < 0.05. Plots and correlation analyses were achieved using Origin 2021 software (OriginLab Corp., Northampton, MA, USA).

## 3. Results

### 3.1. Rice Appearance Quality

Year, variety, N fertilizer dosage, and their interactions significantly affected the chalky grain percentage and chalkiness (Figure 2). The chalky grain percentage and chalkiness were significantly higher in the chalkier variety C1 than in the low-chalky variety C2. The chalky grain percentage and chalkiness of both varieties decreased with increasing N fertilization. After 3-year averaging, the chalky grain percentage and chalkiness were reduced by 8.93–19.33% and 16.84–28.02% in C1 and by 8.61–25.49% and 22.41–33.99% in C2 under the N2 and N3 treatments, respectively, compared to the N1 treatment (Figure 2a,b).

### 3.2. Dynamic Changes in the Activity of Key Enzymes Involved in Grains C Metabolism

The dynamic changes in key enzyme activities of the grain C metabolism in different chalky hybrid indica rice varieties under different N fertilizer dosages are shown in Figure 3. The activities of the key enzymes involved in grain C metabolism differed significantly between varieties and N dosages, and the highest activities mostly occurred 6–30 days after flowering. The activities of the key enzymes of C metabolism were slightly higher in the low-chalky variety C2 than in the highly chalky variety C1, and after sequential averaging by years, days after anthesis, and N treatments, the activities of ADPG, GBSS, SS, BE, and DBE were 36.57%, 17.31%, 39.37%, 16.62%, and 17.77% higher in C2 than in C1 (Figure 3a–j). In addition, the ADPG, SS, and BE activities increased with an increasing N dosage, and vice versa for GBSS and DBE. After averaging over 2 years, compared to N1, C1 and C2 under the N2 and N3 treatments showed increased activity of ADPG (by 21.45–33.59% and 11.15–20.34%), SS (12.76–28.07% and 16.50–31.52%), and BE (8.82–19.74% and 9.16–22.86%), respectively, after 6–42 days of flowering; by contrast, the enzyme activity of GBSS was decreased by 8.18–16.09% and 7.63–16.75% and that of DBE by 7.95–13.27% and 3.34–7.28%, respectively.

### 3.3. Dynamic Changes in the Activity of Key Enzymes Involved in Grains N Metabolism

The dynamic trends of the key enzyme activities in kernel N metabolism are shown in Figure 4. N fertilizer dosage significantly affected the kernel N metabolism key enzyme activities of the different rice varieties, and the C2 N metabolism key enzyme activities were slightly higher than those of C1. With an increasing post-anthesis time, the activities of the key enzymes of N metabolism in the two varieties exhibited a trend of increasing first and then decreasing, followed by leveling off; GDH, GOGAT, and GS peaked at 12–18 days, 6–18 days, and 18–24 days after flowering, respectively (Figure 4a–f). The activities of the key enzymes involved in N metabolism increased with an increasing N application in both varieties (except for GDH in C2). After averaging over 2 years, compared to N1, C1 and C2 showed increased activities of GOGAT (13.73–14.96% and 3.52–18.98%, respectively) and GS (30.01–43.46% and 1.42–9.94%, respectively) after 6–42 days of flowering under the N2 and N3 treatments.

### 3.4. Dynamic Changes in Endogenous Hormones in Grains

The dynamic changes in kernel-endogenous hormones are presented in Figure 5, and their levels were significantly affected by variety and N fertilizer dosage. The levels of kernel-endogenous hormones were slightly higher in C2 than in C1 and higher in 2019 than in 2020. With an increasing post-flowering time, the endogenous hormone levels of both varieties presented a trend of an initial increase followed by a decrease (except for IAA and GAs in C1) and stabilization. The ACC, IAA, GA, ABA, and ZR levels in C2 peaked at 18, 12–18, 12–18, 30, and 18 days after anthesis, respectively, whereas the ACC, ABA, and ZR levels in C1 peaked at 24, 24–30, and 12–18 days after anthesis. Overall, a higher N dosage resulted in higher levels of endogenous hormones in both varieties. After averaging over 2 years, compared to N1, C1 and C2 showed significantly increased levels of ACC (by 21.32–29.84% and 24.06–52.97%, respectively), IAA (by 19.02–35.21% and 11.49–25.06%, respectively), GA (by 27.84–41.52% and 19.10–22.78%, respectively), ABA (by 15.53–43.12% and 19.86–36.60%, respectively), and ZRs (by 8.99–18.10% and 7.97–13.66%, respectively) after 6–42 days of flowering under the N2 and N3 treatments (Figure 5a–j).

### 3.5. Dynamic Changes in Starch and Its Composition in Grains

The dosage of N fertilizer significantly affected the synthesis of starch and its components in the grains of the different rice varieties (Figure 6). Overall, both varieties synthesized and accumulated more starch in 2019 than in 2020, and the C2 kernels exhibited weaker amylose synthesis than the C1 kernels, whereas the opposite was true for amylopectin and total starch. With an increasing post-anthesis time, the accumulation of amylose, amylopectin, and total starch in the grains of both varieties gradually increased and eventually remained stable. Notably, amylose accumulated more slowly, and amylopectin accumulated more rapidly in C2 than in C1 6–12 days after anthesis to promote the rapid accumulation of total starch and reach equilibrium earlier (18 days after anthesis). In both varieties, amylose content showed the order of N1 > N2 > N3 for (Figure 6a,b), which was N1 < N2 < N3 for amylopectin and total starch content (Figure 6c–f).

### 3.6. Dynamic Changes in Free Amino Acids and Total Protein in Grains

The N fertilizer dosage significantly affected the free amino acid and total protein synthesis in the differently chalky rice grains (Figure 7). The accumulation of free amino acids and total protein in grains was higher in 2019 than in 2020 and slightly higher in C2 than in C1. The free amino acids in the grains of both varieties exhibited a single-peak curve with an increasing post-anthesis time, reaching a peak 12 days after anthesis and then decreasing markedly from 12 to 24 days after anthesis, after which, they leveled off. No peak in protein accumulation was observed in either variety, and the rate of synthesis accelerated with an increasing post-anthesis time and decelerated thereafter. After averaging over 2 years, compared to N1, C1 and C2 showed significantly higher levels of free amino acid (by 14.67–15.55% and 4.01–5.42%, respectively) and protein (by 10.28–14.11% and 3.72–13.12%, respectively; Figure 7a–d) after 6–42 days of flowering under the N2 and N3 treatments.

### 3.7. Rice Protein Fractions

The overall performance trends of the rice protein fractions for C1 and C2 were essentially identical in 2019 and 2020 (Figure 8). The albumin, globulin, and prolamin levels were slightly higher in the C2 than in the C1 rice, whereas the opposite trend was observed for glutelin. In addition, under different N fertilizer dosages, the contents of albumin, prolamin, and glutelin in the two rice varieties significantly increased under the N2 and N3 treatments, whereas the globulin content significantly decreased (Figure 8a–d).

### 3.8. Rice Amino Acid Composition

The total amino acid content was significantly higher in the C2 than in the C1 rice and higher in 2019 than in 2020 (Figure 9); the overall performance trend of the amino acid composition was generally similar between years and varieties. In both varieties, Glu, Asp, and Leu accounted for a higher proportion of each amino acid composition type, followed by the rest of the compositions; Among them, the sum of Glu, Asp, and Leu in the C1 and C2 rice accounted for 43.83–48.02% and 47.7–49.94% of the total amino acids under N application, respectively. Also, higher percentages of α-ketoglutarate- and 3-phosphoglycerate-derived amino acids such as Glu, Arg, Ser, Gly, Ala, and Cys were found in the C2 rice compared to C1. The contents of Thr, Asp, Pro, Ile, and Leu in the C1 whole rice decreased with an increasing N dosage, and the contents of Lys, Tyr, and Met increased and then decreased, whereas the remaining amino acids showed an increasing trend. In addition, His, Thr, Asp, Pro, Lys, Tyr, Met, Val, Ile, and Leu in the C2 whole rice decreased with an increase in N fertilizer dosage, whereas the Cys content first increased and then decreased, and the remaining amino acids tended to increase (Figure 9a–d), indicating that an increased N fertilization reduced the synthesis of pyruvate- and oxaloacetate-derived amino acids (Asp, Thr, Lys, Val, Leu, and Ile, etc.), but promoted the accumulation of α-ketoglutarate- and 3-phosphoglycerate-derived types of amino acids (Glu, Arg, Ser, Gly, and Cys, etc.).

### 3.9. Relationship between Rice Chalkiness and Key Enzyme Activities of Grain C and N Metabolism, Endogenous Hormones, and Amino Acid Composition

The chalky grain percentage and chalkiness degree of the two rice varieties were negatively correlated with ADPG, SS, BE, DBE, GOGAT, GS, ACC, IAA, GAs, ABA, ZRs, APC, TC, FAA, PC, AMC, PMC, and GTC, while positively correlated with GBSS, GDH, AC, and GLC (Figure 10a,b). The amino acid composition of rice, chalky grain percentage, and chalkiness degree were negatively correlated with Arg, Glu, Ser, Gly, Ala, Phe, and Try and positively correlated with Thr, Asp, Lys, Tyr, Met, Ile, and Leu (Figure 10c,d).

## 4. Discussion

### 4.1. Responses of Rice Chalkiness to N Fertilization

Chalkiness is a key parameter restricting the commercial value of rice, and N application is an essential cultivation measure that affects chalkiness. Suitable N application contributes to increased rice yields; however, conclusions on its effect on rice chalkiness quality vary [19,39]. In the present study, N application decreased the chalky grain percentage and chalkiness in high- and low-chalky varieties, consistent with the findings of Zhou et al. [26]. Applying N fertilization can optimize the whole process of grain filling and thus reduce the rice chalkiness [20], whereas excessive N application can prolong the grain filling period, reduce the grain filling rate, and cause an increase in chalkiness [17]. Nevertheless, some studies have also concluded that N application increases rice chalkiness [27]. The chalkiness of different varieties responded differently to N fertilizer, and in this study, the chalkiness of high-chalky varieties was more sensitive to N fertilizer compared to low-chalky varieties, which may be related to its changes in amylose and protein from 6–24 days after anthesis. Previous studies have indicated that high temperatures lead to increased rice chalkiness, which can be alleviated by applying a certain amount of N fertilization [39]. Rice chalkiness is positively correlated with temperature at maturity, especially with daily average temperatures after flowering, which is most closely correlated with chalkiness [40]. In this study, by adjusting the sowing date, the heading and flowering dates of the high- and low-chalk7 varieties were basically the same, but the daily average temperature averaged over the 20 days post-flowering period of the two varieties in the three years increased slightly (except in 2021), which, in turn, resulted in an increase in chalkiness, indicating that the daily average temperatures were an important factor influencing the variations in chalk in the high- and low-chalky varieties.

### 4.2. Relationship between Rice Chalkiness Formation and Grain C and N Metabolism

The key enzyme activities of C and N metabolism are the basis for revealing the formation of rice chalkiness; however, how their interaction affects rice chalkiness has not been fully elucidated [16]. In the current study, the key enzyme activities for C and N metabolism in grains were higher in the low-chalky variety, C2. Compared to N1, the N2 and N3 treatments showed higher ADPG, SS, and BE enzyme activities, which increased the synthesis of amylopectin and relied on lower GBSS and DBE enzyme activities to reduce the synthesis of amylose (Figure 3 and Figure 6), thereby supporting a slight increase in total starch in the endosperm cells and reducing the chalky grain percentage and chalkiness. In addition, with an increasing N fertilizer dosage, the activities of the GOGAT, GS, and GDH enzymes increased to varying degrees from 6 to 30 days after flowering, which promoted amino acid metabolism and increased protein synthesis (Figure 4 and Figure 7). When amyloplasts develop normally, the protein is stored in the space between the amyloplasts, which contributes to the tightness of starch grains, thus reducing chalkiness [41]; however, when amyloplasts develop abnormally, more protein accumulates in the grains, which may lead to loose binding between starch and protein bodies, increasing the interstitial space between them, thus increasing chalkiness [42]. Our results also showed that, except for the C metabolism enzyme activity related to the synthesis of amylose, the activities of the key C and N metabolism enzymes were positively correlated with chalkiness, indicating that increasing the N fertilizer effectively maintained the balance of grain C and N metabolism, and increased the C metabolism key enzyme activities while increasing the N metabolism key enzyme activities in synergy, to contribute to the increase in starch and protein in grains (Figure 6 and Figure 7), ultimately improving the appearance and processing quality of rice.

Endogenous hormones in rice grains are involved in the regulation of the activities of key enzymes involved in starch synthesis and indirectly affect starch synthesis, and hence, chalkiness formation [43]. In this study, we found that grain endogenous hormone content had varietal and time accumulation differences, except for ZRs. The low-chalky variety C2 grains endogenous hormone content dominance was obvious, while at 12–24 d after flowering, two varieties of grain hormones ACC, IAA, GAs, and ZRs contents were higher, and ABA was delayed to 24–30 d after flowering (Figure 5), indicating that the activity time of endogenous hormones in different types was different. The active period of endogenous hormones in grains was different under each N fertilizer dose, but increasing N fertilizer increased the content of endogenous hormones in grains, especially ACC, GAs, and ABA, which could help to regulate the maturity of grains and thus affect chalkiness. An imbalance in the metabolism of the endogenous hormone ABA in grains tends to cause a decrease in the activity of starch synthase, an increase in the accumulation of substrate sugars in the sucrose-starch metabolic pathway, and a weakening of the rate of starch synthesis [44]. This may alter the distribution of amylose and amylopectin within the grain, thereby modulating the spatial structure of the starch granules and exacerbating the increase in chalkiness. In the present study, chalkiness was significantly and negatively correlated with grain endogenous hormones (Figure 10), suggesting that increased N fertilization increased grain endogenous hormone synthesis, thus inhibiting soluble sugar and sucrose accumulation and decreasing GBSS enzyme activity and straight-chain starch synthesis; however, it competitively increased ADPG, SS, and BE enzyme activities, thus promoting branched-chain starch synthesis to support a slight increase in the total starch content of the grains and to reduce rice chalkiness.

### 4.3. Relationship between Rice Chalkiness Formation and Secondary Metabolites of Proteins

Rice grains are milled to produce polished rice, which consists mainly of starch, proteins, and fats, and changes in these physicochemical components regulate the formation of rice chalkiness. Higher amounts of proteins and lipids are bound to the surface of starch granules or inserted into the pores of inhomogeneous starch granules [45], forming composite starch granules and enhancing the compactness between starch granules, contribute to a reduction in rice chalkiness, which is consistent with the results of the present study. Cereal storage proteins are classified into albumin, globulin, prolamin, and glutelin based on their solubility. Previous studies have shown that mutations in genes affecting storage protein accumulation can lead to the appearance of a powdery endosperm phenotype in grains [46], whereas an insufficient accumulation of gliadin is highly susceptible to the formation of chalkiness [47]. In the present study, the glutelin, albumin, and prolamin contents in the polished rice of the two varieties increased with increasing N fertilization, whereas the opposite trend was observed for globulin. This indicates that an appropriate increase in N fertilizer increases glutelin, albumin, and prolamin levels and reduces globulin levels, which may help to reduce the chalkiness of rice.

Amino acids are precursors in protein synthesis, and we found that the response trend of the amino acid synthesis in rice of varieties with different chalkiness under different N fertilization schemes converged, and most of the amino acids synthesized by phosphoenolpyruvic acid and oxaloacetic acid derivatives decreased in the polished rice of the high-chalky white variety C1 and the low-chalky white variety C2 as the N fertilizer dosage increased, whereas the content of amino acids synthesized by α-ketoglutaric acid and 3-phosphoglycerolacetic acid derivative was significantly increased. The changes in these amino acids increased the storage proteins in rice grains, especially glutelin, albumin, and prolamin, resulting in a decrease in rice chalkiness, which is slightly different from the study by Xi et al. [15]. This may also be one of the reasons why an increasing N fertilizer application leads to a decrease in rice chalkiness.

## 5. Conclusions

Compared to C1, C2 had stronger activities of the key enzymes involved in grain C and N metabolism, higher endogenous hormones, and greater total amino acids. These physiological parameters are closely associated with a decrease in rice chalkiness following N application. Increased N fertilization boosted grain endogenous hormone content, ADPG, SS, and BE enzyme activities, and synergistically increased GOGAT and GS enzyme activities, but inhibited GBSS and DBE enzyme activities, tending to support the synthesis of amylopectin and α-ketoglutarate and 3-phosphoglyceric acid-derived types of amino acids in the endosperm cells of the grains and optimize the accumulation of starch composition and protein fractions, which ultimately contributed to the increase in starch and protein at 6–24 days post-flowering, in order to sustain the reduction in rice chalkiness and improve the quality of its appearance. Overall, the application of 150 kg of N hm^−1^ improved the appearance quality of both high- and low-chalky varieties.

## Figures and Tables

**Figure 1 foods-13-00855-f001:**
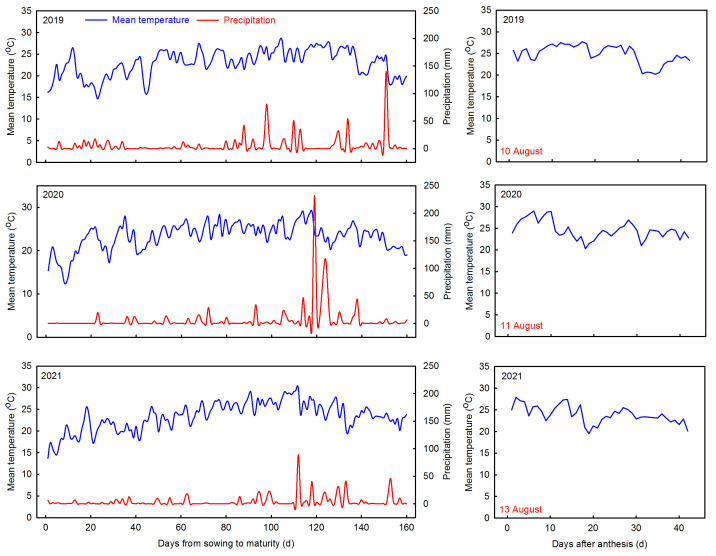
Daily mean temperature and precipitation during the growth period of rice, 2019–2021.

**Figure 2 foods-13-00855-f002:**
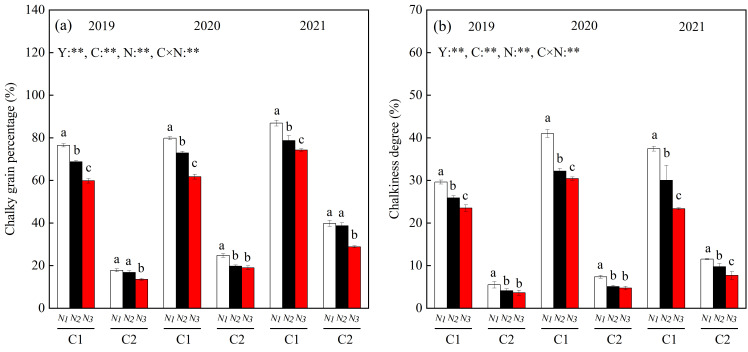
Changes in chalky grain percentage (**a**) and chalkiness (**b**) of rice in different chalky hybrid indica rice varieties under different N fertilizer rates. Different lowercase letters correspond to statistical differences at the 0.05 level for each N application rates in the same year and variety. ** indicates significant at the 0.01 level of probability. Vertical bars represent mean ± standard deviation (*n* = 3). C1, Chuannongyou 508; C2, Shuangyou 573. N1, N2, and N3 denote 75, 150, and 225 kg/hm^2^ of applied N application rates, respectively.

**Figure 3 foods-13-00855-f003:**
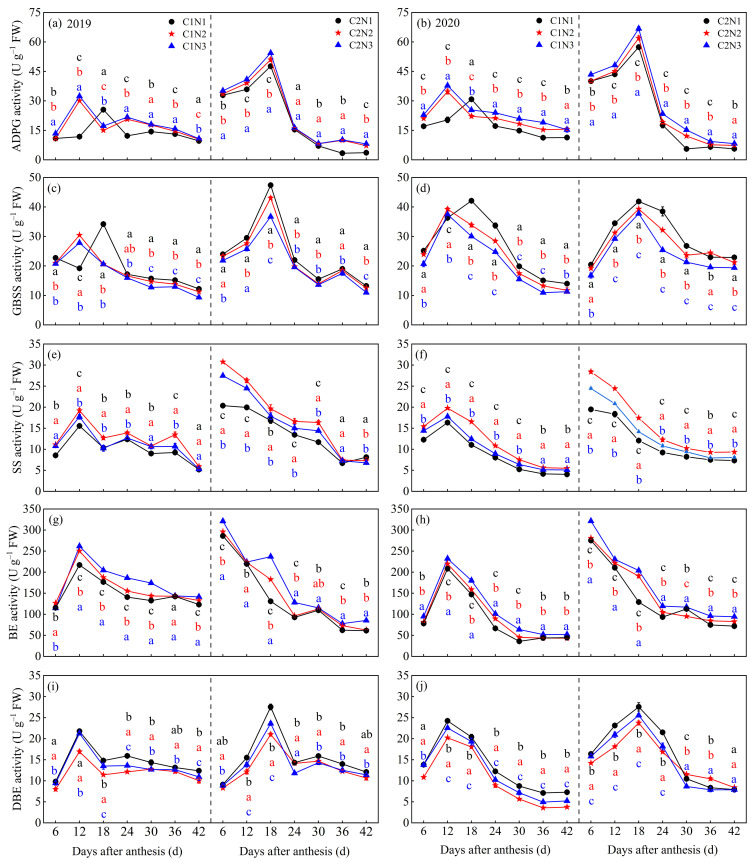
Dynamic changes in the activities of grain ADPG (**a**,**b**), GBSS (**c**,**d**), SS (**e**,**f**), BE (**g**,**h**), and DBE (**i**,**j**) in different chalky hybrid indica rice under different N application rates. Different colored lowercase letters correspond to statistical differences at the 0.05 level for each N application rates in the same year, period and variety. Vertical bars represent mean ± standard deviation (*n* = 3). ADPG, adenosine diphosphate glucose pyrophosphorylase; GBSS, granule-bound starch synthase; SS, soluble starch synthase; BE, starch branching enzyme; and DBE, starch-debranching enzyme. C1, Chuannongyou 508; C2, Shuangyou 573. N1, N2, and N3 denote 75, 150, and 225 kg/hm^2^ of applied N application rates, respectively.

**Figure 4 foods-13-00855-f004:**
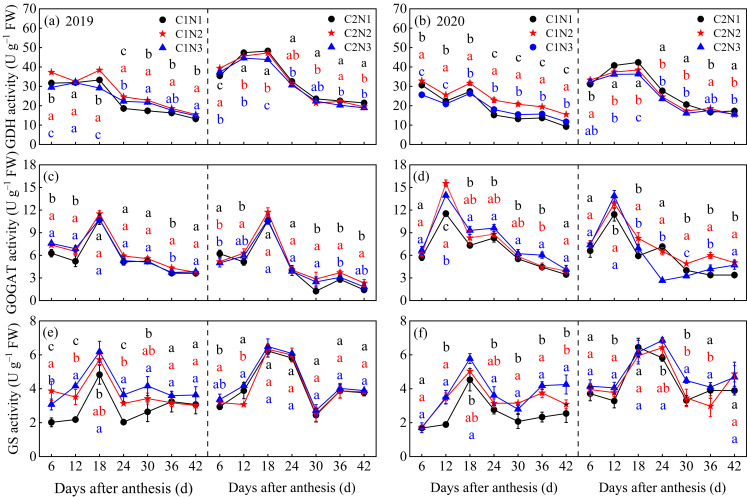
Dynamic changes in the activities of grain GDH (**a**,**b**), GOGAT (**c**,**d**), and GS (**e**,**f**) in different chalky hybrid indica rice under different N application rates. Different colored lowercase letters correspond to statistical differences at the 0.05 level for each N application rates in the same year, period and variety. Vertical bars represent mean ± standard deviation (*n* = 3). GDH, glutamate dehydrogenase; GOGAT, glutamate synthetase; and GS, glutamine synthetase. C1, Chuannongyou 508; C2, Shuangyou 573. N1, N2, and N3 denote 75, 150, and 225 kg/hm^2^ of applied N application rates, respectively.

**Figure 5 foods-13-00855-f005:**
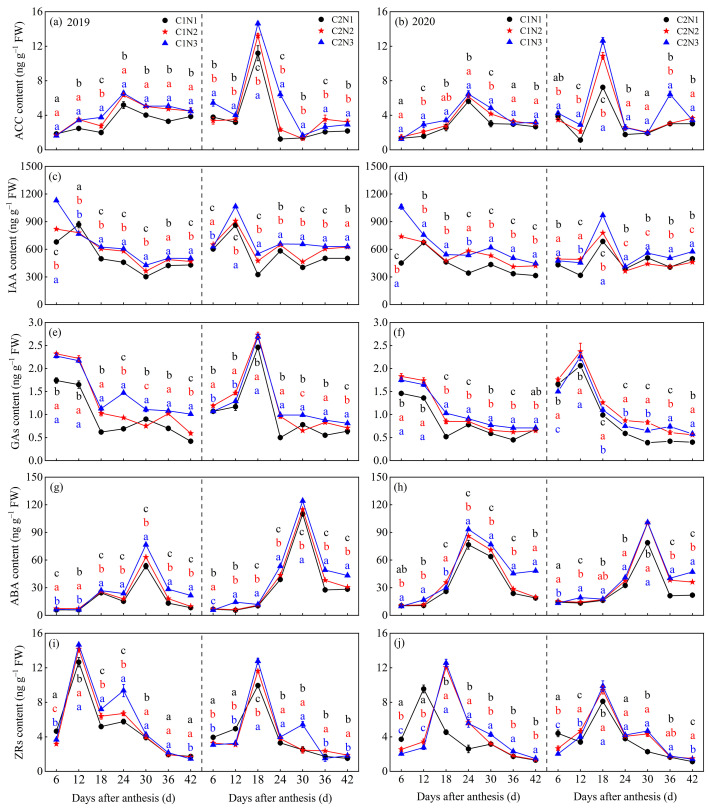
Dynamic changes of endogenous hormones ACC (**a**,**b**), IAA (**c**,**d**), GAs (**e**,**f**), ABA (**g**,**h**), and ZRs (**i**,**j**) in grains of different chalky hybrid indica rice under different N application rates. Different colored lower case letters correspond to statistical differences at the 0.05 level for each N application rates in the same year, period and variety. Vertical bars represent mean ± standard deviation (*n* = 3). ACC, 1-aminocyclopropane 1-carboxylic acid; ABA, abscisic acid; IAA, indoleacetic acid; GAs, gibberellins; and ZRs, cytokinins. C1, Chuannongyou 508; C2, Shuangyou 573. N1, N2 and N3 denote 75, 150, and 225 kg/hm^2^ of applied N application rates, respectively.

**Figure 6 foods-13-00855-f006:**
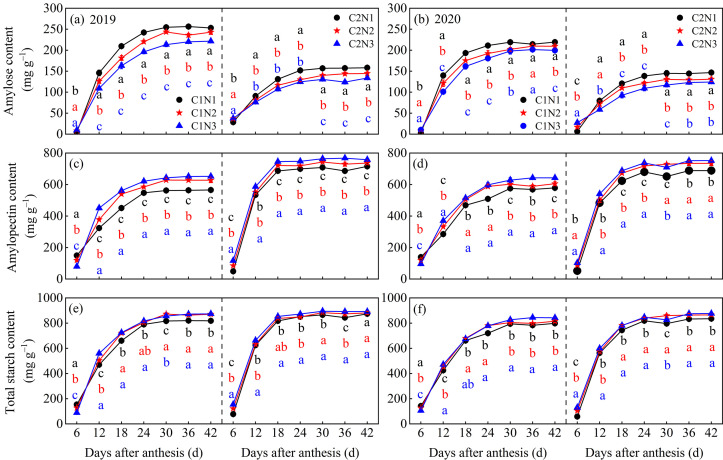
Dynamic changes of amylose (**a**,**b**), amylopectin (**c**,**d**), and total starch (**e**,**f**) in grains of different chalky hybrid indica rice under different nitrogen application rates. Different colored lowercase letters correspond to statistical differences at the 0.05 level for each N application rates in the same year, period and variety. Vertical bars represent mean ± standard deviation (*n* = 3). C1, Chuannongyou 508. C2, Shuangyou 573. N1, N2, and N3 denote 75, 150, and 225 kg/hm^2^ of applied N application rates, respectively.

**Figure 7 foods-13-00855-f007:**
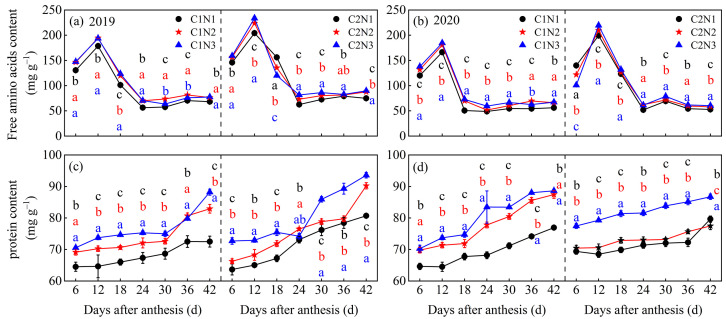
Dynamic changes in free amino acids (**a**,**b**) and total protein (**c**,**d**) in grains of different chalky hybrid indica rice under different N application rates. Different colored lowercase letters correspond to statistical differences at the 0.05 level for each N application rates in the same year, period and variety. Vertical bars represent mean ± standard deviation (*n* = 3). C1, Chuannongyou 508. C2, Shuangyou 573. N1, N2, and N3 denote 75, 150, and 225 kg/hm^2^ of applied N application rates, respectively.

**Figure 8 foods-13-00855-f008:**
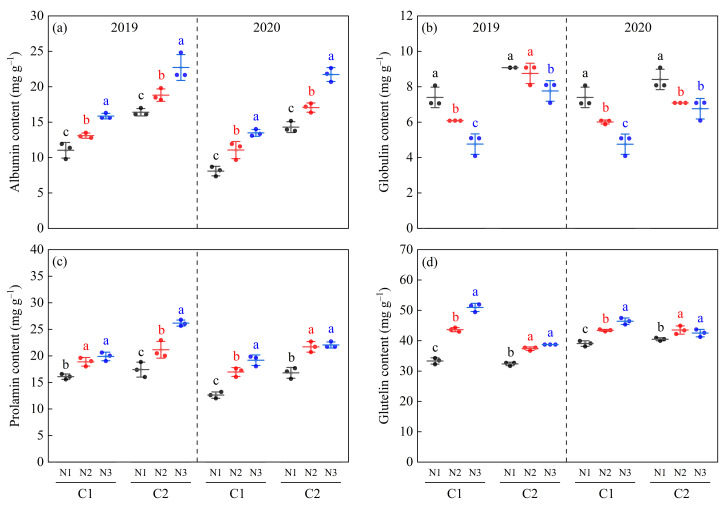
Effect of N application rates on rice albumin (**a**), globulin (**b**), prolamin (**c**), and glutelin (**d**) of different chalky hybrid indica rice. Different colored lowercase letters correspond to statistical differences at the 0.05 level for each N application rates in the same year, period and variety. C1, Chuannongyou 508. C2, Shuangyou 573. N1, N2, and N3 denote 75, 150, and 225 kg/hm^2^ of applied N application rates, respectively.

**Figure 9 foods-13-00855-f009:**
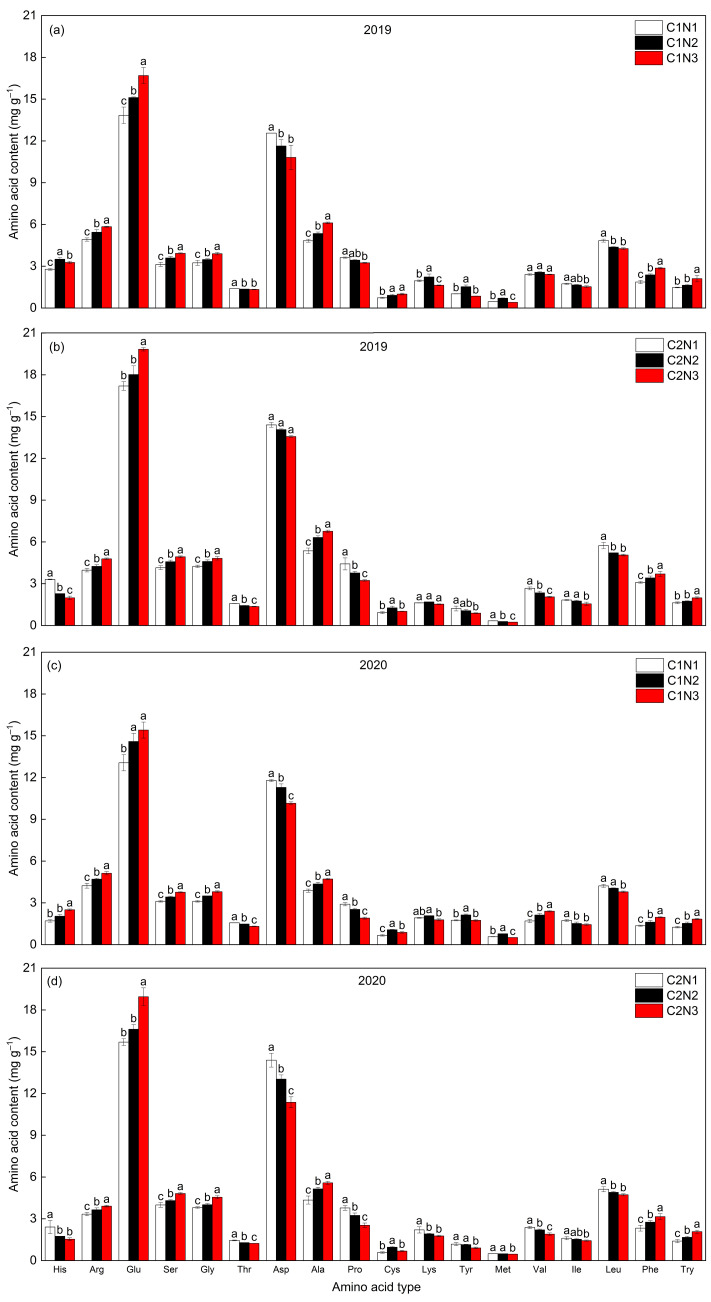
Effect of N application rates on the content of amino acid components (**a**–**d**) in different chalky hybrid indica rice. Different lowercase letters correspond to statistical differences at the 0.05 level for each N application rates in the same year, period and variety. Vertical bars represent mean ± standard deviation (*n* = 3). His, histidine; Arg, arginine; Glu, glutamic acid; Ser, serine; Gly, glycine; Thr, threonine; Asp, aspartic acid; Ala, alanine; Pro, proline; Cys, cysteine; Lys, lysine; Tyr, tyrosine; Met, methionine; Val, valine; Ile, isoleucine; Leu, leucine; Phe, phenylalanine; and Try, tryptophan. C1, Chuannongyou 508; C2, Shuangyou 573. N1, N2, and N3 denote 75, 150, and 225 kg/hm^2^ of applied N application rates, respectively.

**Figure 10 foods-13-00855-f010:**
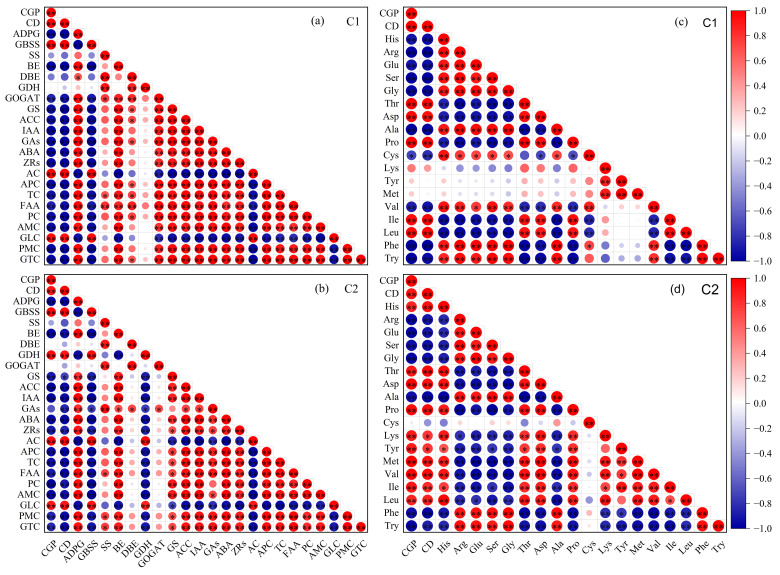
Relationship between rice chalkiness and key enzyme activities of grain C and N metabolism, endogenous hormones (**a**,**b**) and amino acid composition (**c**,**d**). C1, Chuannongyou 508. C2, Shuangyou 573. CGP, chalky grain percentage; CD, chalkiness degree; ADPG, adenosine diphosphate glucose pyrophosphorylase; GBSS, granule-bound starch synthase; SS, soluble starch synthase; BE, starch branching enzyme; DBE, starch-debranching enzyme; GDH, glutamate dehydrogenase; GOGAT, glutamate synthetase; GS, glutamine synthetase; ACC, 1-aminocyclopropane 1-carboxylic acid; ABA, abscisic acid; IAA, indoleacetic acid; GAs, gibberellins; ZRs, cytokinins; AC, amylose; APC, amylopectin; TC, total starch; FAA, free amino acid; PC, protein; AMC, albumin; GLC, globulin; PMC, prolamin; GTC, glutelin; His, histidine; Arg, arginine; Glu, glutamic acid; Ser, serine; Gly, glycine; Thr, threonine; Asp, aspartic acid; Ala, alanine; Pro, proline; Cys, cysteine; Lys, lysine; Tyr, tyrosine; Met, methionine; Val, valine; Ile, isoleucine; Leu, leucine; Phe, phenylalanine; and Try, tryptophan. * and **, significant at the 0.05 and 0.01 probability levels, respectively.

**Table 1 foods-13-00855-t001:** Basic fertility of tillage soil (0–20 cm) of experimental field, 2019–2021.

Year	pH Value	P (mg kg^−1^)	K (mg kg^−1^)	N (mg kg^−1^)	Organic Matter (g kg^−1^)	Total N (g kg^−1^)
2019	6.55	28.49	86.32	101.31	21.30	1.53
2020	6.76	28.15	83.33	92.75	19.76	1.32
2021	6.74	26.90	82.60	95.81	19.83	1.45

## Data Availability

The original contributions presented in the study are included in the article or Appendix A, further inquiries can be directed to the corresponding author.

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
