# Peer review of "Grain Chalkiness Is Decreased by Balancing the Synthesis of Protein and Starch in Hybrid Indica Rice Grains under Nitrogen Fertilization"

_foods, 2024, doi:10.3390/foods13060855_

Round 1
Reviewer 1 Report
Comments and Suggestions for Authors
This manuscript demonstrates the effect of N fertilization on rice grain chalkiness, starch synthesis enzymes, N assimilation enzymes, plant hormones, starch/storage protein/free amino acids compositions using two hybrid indica rice lines with high and low grain chalkiness over three years field experiments. The authors concluded that N fertilization can decrease rice grain chalkiness by balancing grain protein and starch composition. The data demonstrated in this manuscript look interesting and important, however no direct evidence was shown but only descriptive explanation was indicated to support the importance of nitrogen-carbon balance. The description that is most difficult to understand is, for example, that ‘Under excessive N application, the grain filling rate are accelerated (lines 497-500)’. I think the excess N application prolongs a grain filling period and reduces the grain filling rate. The authors are advised to show evidence or refer good literatures to support their ideas. The other comments are shown below.
1. The rice grain chalkiness is affected by both genetic and environmental factors. The authors are advised to differentially discuss the two factors. I speculate the high (C1) and low (C2) chalkiness lines are more suffered from genetic and environmental factors, respectively. There was almost no description of the effect of high air temperature during the grain filling period, which is more important environmental factor than N fertilization, on the grain chalkiness. It might be interesting to see the difference in the response to air temperature (especially temperature during 20 days after flowering) between the two lines. Please indicate the flowering date in Fig. 1 and show mean air temperature during 20 days after flowering.
2. Most enzyme activities and plant hormone content were shown at fresh weight basis. I am wondering how the grain fresh weight was differed between N treatments, hybrid lines, and grain filling stages because that largely affects the data at fresh weight basis. Can the authors show the grain fresh weigh data?
Minor comments are shown below.
3. Gliadin and glutenin (gluten) are storage proteins of wheat. The words prolamin and glutelin should be used.
4. Line 20; free amino acid composition
5. Line 43; gelatin?
6. Line 52; stacked? packed?
7. Line 75; grain filling node rate?
8. Line 177; a sieve ton conduct dynamic determination, please add more explanation
9. Line 192; were the centrifuged
10. Line 272; 3 kg paddy?
11. Line 365; C2 than in C1
12. Lines 444-448; first increased and then decreased, what do you mean?
13. Figure 9; Bar chart is better than line chart because there is no relationship between amino acids.
Author Response
Response to reviewers' comments
We appreciate the positive comments and constructive suggestions by the reviewers. We have addressed all comments, and the detailed point-by-point responses to the comments are provided below. Our responses to the reviewers’ comments are shown below in blue. Any new or revised text added to the manuscript is shown below in red.
Reviewer#1: This manuscript demonstrates the effect of N fertilization on rice grain chalkiness, starch synthesis enzymes, N assimilation enzymes, plant hormones, starch/storage protein/free amino acids compositions using two hybrid indica rice lines with high and low grain chalkiness over three years field experiments. The authors concluded that N fertilization can decrease rice grain chalkiness by balancing grain protein and starch composition. The data demonstrated in this manuscript look interesting and important, however no direct evidence was shown but only descriptive explanation was indicated to support the importance of nitrogen-carbon balance. The description that is most difficult to understand is, for example, that‘Under excessive N application, the grain filling rate are accelerated (lines 497-500)’. I think the excess N application prolongs a grain filling period and reduces the grain filling rate. The authors are advised to show evidence or refer good literatures to support their ideas. The other comments are shown below.
Response: We thank the reviewer for this valuable suggestion. We have made some revisions and added references (lines 498-500, 679-680).
“In the present study, N application decreased chalky grain percentage and chalkiness in high and low chalky varieties, consistent with the findings of Zhou et al. [26].
- Guo, C.; Yuan, X.; Yan, F.; Xiang, K.; Wu, Y.; Zhang, Q.; Wang, Z.; He, L.; Fan, P.; Yang, Z.; Chen, Z.; Sun, Y.; Ma, J. Nitrogen application rate affects the accumulation of carbohydrates in functional leaves and grains to improve grain filling and reduce the occurrence of chalkiness. Plant Sci. 2022, 13, 921130.
- Zhou, L.; Liang, S.; Ponce, K.; Marundon, S.; Ye, G.; Zhao, X. Factors affecting head rice yield and chalkiness in indica rice. Field Crops Res. 2015, 172, 1–10.
- Idowu, O.; Katsube-Tanaka, T.; Shiraiwa, T. Nitrogen fertilizer application does not always improve available carbohydrate per spikelet but decreases chalkiness under high temperature in rice (Oryza sativa L.) grains.Field Crops Res. 2023, 290.”
1. The rice grain chalkiness is affected by both genetic and environmental factors. The authors are advised to differentially discuss the two factors. I speculate the high (C1) and low (C2) chalkiness lines are more suffered from genetic and environmental factors, respectively. There was almost no description of the effect of high air temperature during the grain filling period, which is more important environmental factor than N fertilization, on the grain chalkiness. It might be interesting to see the difference in the response to air temperature (especially temperature during 20 days after flowering) between the two lines. Please indicate the flowering date in Fig. 1 and show mean air temperature during 20 days after flowering.
Response: We thank the reviewer for this valuable suggestion. As we have adjusted the sowing dates of the two varieties so that they have essentially the same heading and flowering dates (removing temperature disturbances), However, the differences in daily average temperature between the varieties themselves and between years have not been taken into account. Therefore, we have supplemented them. Meanwhile, we have indicated the flowering date in Figure 1 and displayed the daily average temperature after flowering (lines 150, 504-517, 708-709).
“The chalkiness of different varieties responded differently to N fertilizer, and in this study, the chalkiness of high chalky varieties was more sensitive to N fertilizer compared to low chalky varieties, which may be related to its changes in amylose and protein from 6-24 days after anthesis. Previous studies have indicated that high temperatures lead to increased rice chalkiness, which can be alleviated by applying a certain amount of N fertilization [39]. Rice chalkiness is positively correlated with temperature at maturity, especially with the daily average temperatures after flowering is most closely correlated with chalkiness [40]. In this study, by adjusting the sowing date, the heading and flowering dates of the high and low chalk varieties were basically the same, but the daily average temperature averaged over the 20 days post-flowering period of the two varieties in the three years increased slightly (except in 2021), which in turn resulted in an increase in chalkiness, indicating that the daily average temperatures were an important factor influencing the variation of chalk in the high and low chalk varieties.
- Chen, C.; Huang, J.L.; Zhu, L.Y .; Shah, F.; Nie, L.X.; Cui, K.H.; Peng, S.B. Varietal difference in the response of rice chalkiness to temperature during ripening phase across different sowing dates. Field Crops Res. 2013, 151, 85–91.”
2. Most enzyme activities and plant hormone content were shown at fresh weight basis. I am wondering how the grain fresh weight was differed between N treatments, hybrid lines, and grain filling stages because that largely affects the data at fresh weight basis. Can the authors show the grain fresh weigh data?
Response: Thank you for your comment. We apologize for the unclear description. The enzyme activity of both varieties at different stages was measured using a 0.1 g sample, while the grain hormone was measured using a 0.1 mg sample. We have revised it. The specific graphing data has been provided in the supplementary file (lines 194-195).
“Twenty-five uniformly sized kernels were shelled, weighed 0.1 g as a sample, ground in 4.0 ml of extract (different for each enzyme) and then centrifuged at 11 000 × g for 20 min”
Minor comments are shown below.
3. Gliadin and glutenin (gluten) are storage proteins of wheat. The words prolamin and glutelin should be used.
Response: We thank the reviewer for this valuable suggestion. We have revised“Gliadin” to “prolamin” and “glutenin” to “glutelin” (lines 105,108, 283, etc.).
4. Line 20; free amino acid composition
Response: Thank you for your comment. We have revised“free amino acid composition” to “amino acid” (line 23).
5. Line 43; gelatin?
Response: Thank you for your comment. We have revised“gelatin” to “gel” (line47).
6. Line 52; stacked? packed?
Response: Thank you for your comment. We have revised“stacked” to “arranged” (line 56).
7. Line 75; grain filling node rate?
Response: Thank you for your comment. We have changed “grain filling node rate” to “grain filling balance” (line 79).
8. Line 177; a sieve ton conduct dynamic determination, please add more explanation
Response: Thank you for your comment, We add a more detailed description (lines 178-180).
“ hulled, and pulverised in a pulveriser, after which they were passed through a 100-mesh sieve to conduct dynamic determination......
9. Line 192; were the centrifuged
Response: Thank you for your comment. We have revised it (line 195).
“.......and then centrifuged at 11 000 × g for 20 min,....”
10. Line 272; 3 kg paddy?
Response: Thank you for your comment. We have revised“3 kg paddy” to “3 kg of rice” (line 276).
11. Line 365; C2 than in C1
Response: Thank you for your comment. We have revised“C1 than in C1” to “C2 than in C1” (line 370).
12. Lines 444-448; first increased and then decreased, what do you mean?
Response: Thank you for your comment. We want to express that Lys, Tyr and Met showed an increasing and then decreasing trend with increasing N application. We have revised it (lines 451-453).
“The contents of Thr, Asp, Pro, Ile and Leu in C1 whole rice decreased with increasing N dosage, and the contents of Lys, Tyr and Met increased and then decreased, .......”
13. Figure 9; Bar chart is better than line chart because there is no relationship between amino acids.
Response: We thank the reviewer for this valuable suggestion. We have amended Figure 9 to a bar chart.
Reviewer 2 Report
Comments and Suggestions for Authors
Dear Editors and Authors,
I read with interest the manuscript entitled “Grain chalkiness is decreased by balancing the synthesis of protein and starch in hybrid indica rice grains under nitrogen fertilization”. In this study, we conducted a three-year field experiment on hybrid indica rice varieties with varying chalkiness which were subjected to three N fertilization schemes in order (i) to test the effect of N application on the chalky characteristics of different rice varieties; (ii) to assess the accumulation of grain assimilates, endogenous hormones, and activities of key enzymes of C and N metabolism, as well as the relationship between rice starch, protein, and amino acid composition and chalkiness formation; and (iii) to systematically analyze the response mechanism of rice chalkiness to N fertilization. The article's subject is important and has great relevance for the scientific environment of the study area. Therefore, the manuscript needs some adjustments so that it can then be forwarded to the publication process. The manuscript needs the following adjustments:
ABSTRACT
- Detail the methodology part in more detail. There is little information about the methods used. The authors further detailed the results. To review.
- Replace repeated keywords in the title.
INTRODUCTION
- Add a paragraph about the importance of nitrogen fertilization. Cite similar studies carried out previously.
- Add hypotheses, before objectives.
MATERIAL AND METHODS
- In Table 1, you do not need to mention “Available”. Mention only the nutrient, “P”, “N”, ....
- In Figure 1, remove the points, leaving only the lines.
- The application of K2O and P2O5 was based on whose recommendation? Is this a recommendation for China? To quote. This information is important and needs citation.
- All analyses carried out require references. Some are missing.
RESULTS
- In Figure 3, Figure, Figure 5, and Figure 6, add the average error at each point.
- Modify the colors of the correlation figures. Observing the significance of the points is impossible due to this difficult-to-visualize color.
DISCUSSION
-Review this entire section. Some information is a literature review and is not related to the main results found in this work.
CONCLUSIONS
- Add the concentration of N that provided the best results.
Author Response
Response to reviewers' comments
We appreciate the positive comments and constructive suggestions by the reviewers. We have addressed all comments, and the detailed point-by-point responses to the comments are provided below. Our responses to the reviewers’ comments are shown below in blue. Any new or revised text added to the manuscript is shown below in red.
Reviewers#2: I read with interest the manuscript entitled “Grain chalkiness is decreased by balancing the synthesis of protein and starch in hybrid indica rice grains under nitrogen fertilization”. In this study, we conducted a three-year field experiment on hybrid indica rice varieties with varying chalkiness which were subjected to three N fertilization schemes in order (i) to test the effect of N application on the chalky characteristics of different rice varieties; (ii) to assess the accumulation of grain assimilates, endogenous hormones, and activities of key enzymes of C and N metabolism, as well as the relationship between rice starch, protein, and amino acid composition and chalkiness formation; and (iii) to systematically analyze the response mechanism of rice chalkiness to N fertilization. The article's subject is important and has great relevance for the scientific environment of the study area. Therefore, the manuscript needs some adjustments so that it can then be forwarded to the publication process. The manuscript needs the following adjustments:
ABSTRACT
-Detail the methodology part in more detail. There is little information about the methods used. The authors further detailed the results. To review.
Response: We thank the reviewer for this valuable suggestion. We have further elaborated on the methods and results section (lines 20-24, 28-38).
“The 2019, 2020 and 2021 trials were conducted in a split-plot design, with high and low chalky varieties as the main plot and N fertilizer rate as the split-plot. The effects of fertilization with 75, 150, and 225 kg N ha-1 on the dynamic synthesis of starch and protein and endogenous hormones and on the amino acid composition of hybrid indica rice kernels with different degrees of chalkiness was investigated.”
“Increased N fertilization decreased the activities of granule-bound starch synthase and starch debranching enzyme, but significantly increased adenosine diphosphate glucose pyrophosphorylase, soluble starch synthase, and starch branching enzyme activities, and synergistically improved glutamate synthetase and glutamine synthetase enzyme activities, which tended to support synthesis of amylopectin, α-ketoglutarate, and 3-phosphoglyceric acid-derived amino acids in the endosperm cells of the grains; this favoured starch and protein accumulation in the grains at 6–30 days after anthesis. Additionally, N application promoted the synthesis of endogenous hormones 1-aminocyclopropane-1-carboxylic acid, gibberellins, and abscisic acid in grains. Hence, N fertilization reduced rice chalkiness in hybrid indica rice varieties by balancing grain protein and starch composition and enhancing some endogenous hormone synthesis.”
-Replace repeated keywords in the title.
Response: We thank the reviewer for this valuable suggestion. We have replaced the duplicate keywords (lines 39-40).
“N rate; Chalkiness features; C and N metabolism; Dynamic accumulation”
INTRODUCTION
-Add a paragraph about the importance of nitrogen fertilization. Cite similar studies carried out previously.
Response: We thank the reviewer for this valuable suggestion. We have added this section (lines 112-122).
“N is an essential nutrient for rice growth, an important component of cellular nucleic acids, endogenous hormones, phospholipids, and proteins, and plays an irreplaceable role in its growth process. Appropriate N fertilizer application is conducive to maintaining the balance between C and N metabolism in rice plants and smooth grain filling, which reduces rice chalkiness [26], whereas excessive N abundance and N deficiency lead to increased chalkiness [27]. Under N deficiency, rice plants exhibit poor nutrient supply, slower material transport and synthesis, insufficient grain filling, an increased number of empty grains, and an increased chalky grain rate and chalkiness. Under excessive N application, nutrient synthesis in rice plants and the grain filling rate are accelerated and grain filling fluctuates, which is not conducive to the accumulation of assimilates and grain filling and is more likely to cause increased chalkiness [15,16].”
-Add hypotheses, before objectives.
Response: We thank the reviewer for this valuable suggestion. We have added the hypotheses (lines 127-130).
“ We thus hypothesized that rice chalkiness may be effectively reduced by increasing endogenous hormones, activities of key enzymes in C and N metabolism, and amino acid interactions in grains, thus balancing the biosynthesis of protein and starch under appropriate N fertilization.”
MATERIAL AND METHODS
-In Table 1, you do not need to mention “Available”. Mention only the nutrient, “P”, “N”, ....
Response: We thank the reviewer for this comment. In Table 1, we have removed "available" (line 150).
-In Figure 1, remove the points, leaving only the lines.
Response: We thank the reviewer for this comment. In Figure 1, we have removed the points, leaving only the lines (line 150).
-The application of K2O and P2O5 was based on whose recommendation? Is this a recommendation for China? To quote. This information is important and needs citation.
Response: We thank the reviewer for this valuable suggestion. We have added references (lines 683-684).
“28. Yang, Z.; Li, N.; Ma, J.; Sun, Y.; Xu, H. High-yielding traits of heavy panicle varieties under triangle planting geometry: a new plant spatial configuration for hybrid rice in china. Field Crop Res. 2014, 168, 135–147.”
-All analyses carried out require references. Some are missing.
Response: We thank the reviewer for this comment. We have checked this section and added the missing references(lines 683-686, 695-700).
“28. Yang, Z.; Li, N.; Ma, J.; Sun, Y.; Xu, H. High-yielding traits of heavy panicle varieties under triangle planting geometry: a new plant spatial configuration for hybrid rice in china. Field Crop Res. 2014, 168, 135–147.
- Chen, Z.; Li, P.; Du, Y.; Jiang, Y.; Cai, M.; Cao, C. Dry cultivation and cultivar affect starch synthesis and traits to define rice grain quality in various panicle parts. Carbohydrate Polym. 2021, 269, 118336.
- Balcke, G.; Handrick, V.; Bergau, N.; Fichtner, M.; Henning, A.; Stellmach, H.; Tissier, A.; Hause, B.; Frolov, A. An UPLC-MS/MS method for highly sensitive high-throughput analysis of phytohormones in plant tissues. Plant Methods. 2012, 8, 47.
- Glauser, G.; Grund, B.; Gassner, A.; Menin, L.; Henry, H.; Bromirski, M.; Schutz, F.; McMullen, J.; Rochat, B. Validation of the mass-extraction-window for quantitative methods using liquid chromatography high resolution mass spectrometry. Anal chem. 2016, 88, 3264–3271.”
RESULTS
-In Figure 3, Figure 4, Figure 5, and Figure 6, add the average error at each point.
Response: We thank the reviewer for this valuable suggestion. Figures 3, 4, 5, and 6 already contain average errors, but the average errors are small and were masked when we made the graphs, and we have submitted the specific graphing data to the Supplementary file.
-Modify the colors of the correlation figures. Observing the significance of the points is impossible due to this difficult-to-visualize color.
Response: We have increased the size and deepened the colours of the point and line diagrams for easier viewing, and made some modifications to the related graphics, e.g. Figures 1, 2 and 9.
DISCUSSION
-Review this entire section. Some information is a literature review and is not related to the main results found in this work.
Response: We thank the reviewer for this comment. We checked the discussion section, removed the literature review section that was not relevant to the main results and made some revisions (lines 498-517, 579-580).
“In the present study, N application decreased chalky grain percentage and chalkiness in high and low chalky varieties, consistent with the findings of Zhou et al. [26]. Appropriate N application shortened the grain filling period, promoted grain filling and led to a reduction in chalkiness [19], whereas excessive N application prolonged the grain filling period, reduced the grain filling rate and caused an increase in chalkiness [17]. Nevertheless, some studies have also concluded that N application increases rice chalkiness [27]. The chalkiness of different varieties responded differently to N fertilizer, and in this study, the chalkiness of high chalky varieties was more sensitive to N fertilizer compared to low chalky varieties, which may be related to its changes in amylose and protein from 6-24 days after anthesis. Previous studies have indicated that high temperatures lead to increased rice chalkiness, which can be alleviated by applying a certain amount of N fertilization [39]. Rice chalkiness is positively correlated with temperature at maturity, especially with the daily average temperatures after flowering is most closely correlated with chalkiness [40]. In this study, by adjusting the sowing date, the heading and flowering dates of the high and low chalk varieties were basically the same, but the daily average temperature averaged over the 20 days post-flowering period of the two varieties in the three years increased slightly (except in 2021), which in turn resulted in an increase in chalkiness, indicating that the daily average temperatures were an important factor influencing the variation of chalk in the high and low chalk varieties.”
CONCLUSIONS
-Add the concentration of N that provided the best results.
Response: We thank the reviewer for this valuable suggestion. We have added the N concentration that can achieve the best results (lines 607-608).
“Overall, application of 150 kg N hm-1 improved the appearance quality of both high and low chalky varieties.”
Round 2
Reviewer 1 Report
Comments and Suggestions for Authors
The reviewer agreed that the manuscript was properly revised in most parts. However, the following issue is still unclear.
1.
According to my major comment, the authors revised the manuscript and described ‘Appropriate N application shortened the grain filling period, promoted grain filling and led to a reduction in chalkiness [19]’ in Lines 485-486. However, I think this description is still misleading. The appropriate N application (probably the authors referred N2: 150 kg ha−1 and N3: 225 kg ha−1 treatments) showed low chalkiness but did not shorten the grain filling period compared to the N1: 75 kg ha−1 treatment in the reference [19].
[19] Guo, C.; Yuan, X.; Yan, F.; Xiang, K.; Wu, Y.; Zhang, Q.; Wang, Z.; He, L.; Fan, P.; Yang, Z.; Chen, Z.; Sun, Y.; Ma, J. Nitrogen application rate affects the accumulation of carbohydrates in functional leaves and grains to improve grain filling and reduce the occurrence of chalkiness. Front. Plant Sci. 2022, 13, 921130.
Author Response
Response to reviewers' comments
We appreciate the positive comments and constructive suggestions by the reviewers. We have addressed all comments, and the detailed point-by-point responses to the comments are provided below. Our responses to the reviewers’ comments are shown below in blue. Any new or revised text added to the manuscript is shown below in red.
Reviewer#1: According to my major comment, the authors revised the manuscript and described ‘Appropriate N application shortened the grain filling period, promoted grain filling and led to a reduction in chalkiness [19]’ in Lines 485-486. However, I think this description is still misleading. The appropriate N application (probably the authors referred N2: 150 kg ha−1 and N3: 225 kg ha−1 treatments) showed low chalkiness but did not shorten the grain filling period compared to the N1: 75 kg ha−1 treatment in the reference [19].
[19] Guo, C.; Yuan, X.; Yan, F.; Xiang, K.; Wu, Y.; Zhang, Q.; Wang, Z.; He, L.; Fan, P.; Yang, Z.; Chen, Z.; Sun, Y.; Ma, J. Nitrogen application rate affects the accumulation of carbohydrates in functional leaves and grains to improve grain filling and reduce the occurrence of chalkiness. Front. Plant Sci. 2022, 13, 921130.
Response: We thank the reviewer for this valuable suggestion. I have modified it to "Applying N fertilization can optimize the whole process of grain filling and thus reduce the rice chalkiness [20]" (lines 500-501).
[20] Zhang, J.; Zhang, Y.; Song, N.; Chen, Q.; Sun, H.; Peng, T.; Huang, S.; Zhao, Q. Response of grain-filling rate and grain quality of mid-season indica rice to nitrogen application. J. Integr. Agric. 2021, 20, 1465-1473.
Reviewer 2 Report
Comments and Suggestions for Authors
Dear,
The authors made the suggestions proposed in the previous version. Therefore, the article has the potential to be published in this newspaper.
Author Response
We thank you for your review.